# Everything is Context: Agentic File System Abstraction for Context Engineering

a
institution
city, state, country

b
institution
city, state, country

c
institution
city, state, country

d
institution
city, state, country

e
institution
city, state, country

f
institution
city, state, country

## Abstract

Generative AI (GenAI) has reshaped software system design by introducing foundation models as pre-trained subsystems that redefine architectures and operations. The emerging challenge is no longer model fine-tuning but context engineering-how systems capture, structure, and govern external knowledge, memory, tools, and human input to enable trustworthy reasoning. Existing practices such as prompt engineering, retrieval-augmented generation (RAG), and tool integration remain fragmented, producing transient artefacts that limit traceability and accountability. This paper proposes a file-system abstraction for context engineering, inspired by the Unix notion that "everything is a file". The abstraction offers a persistent, governed infrastructure for managing heterogeneous context artefacts through uniform mounting, metadata, and access control. Implemented within the open-source AIGNE framework, the architecture realises a verifiable context-engineering pipeline, comprising the Context Constructor, Loader, and Evaluator, that assembles, delivers, and validates context under token constraints. As GenAI becomes an active collaborator in decision support, humans play a central role as curators, verifiers, and co-reasoners. The proposed architecture establishes a reusable foundation for accountable and human-centred AI co-work, demonstrated through two exemplars: an agent with memory and an MCP-based GitHub assistant. The implementation within the AIGNE framework demonstrates how the architecture can be operationalised in developer and industrial settings, supporting verifiable, maintainable, and industry-ready GenAI systems.

***CCS Concepts:*** • **Software and its engineering**; • **Computing methodologies → Artificial intelligence**;

***Keywords:*** AI engineering, context engineering, GenAI

**ACM Reference Format:**
a, b, c, d, e, and f. 2018. Everything is Context: Agentic File System Abstraction for Context Engineering . In *Proceedings of (AIWare'26)*. ACM, New York, NY, USA, 9 pages. https://doi.org/XXXXXXX.XXXXXXX

## 1 Introduction

Context engineering is emerging as a central concern in software architecture for Generative AI (GenAI) and Agentic systems[4, 12, 18]. It refers to the process of capturing, structuring, and governing external knowledge, memory, tools, and human input so that reasoning by large language models (LLMs) and agents is grounded in the right information, constraints, and provenance. In contrast to prompt engineering, which focuses on crafting individual instructions, context engineering focuses on the entire information lifecycle, from selection, retrieval, filtering, construction, to compression, evaluation and refresh, ensuring that GenAI systems and agents remain coherent, efficient, and verifiable over time.

Recent industrial frameworks such as LangChain have begun to articulate context engineering as a structured process within agent architectures [12]. Four key stages of context engineering are identified. Agents first *write* contextual information into a shared memory or store, *select* the most relevant elements for a given task, *compress* the selected context to fit model constraints, and *isolate* the final subset across agents for reasoning. These industrial practices highlight the growing recognition that context management has become a central architectural concern. Similar pipelines appear in AutoGen [19], and other related frameworks that support tool use and memory augmentation [3]. However, these solutions remain ad hoc and implementation-driven. They lack a unified architectural foundation to ensure traceability,

governance, or systematic handling of evolving context. Consequently, context artefacts generated during these steps are often transient, opaque, and unverifiable, raising challenges of context rot [22] and knowledge drift in industrial-setting GenAI systems [11].

GenAI introduces new architectural constraints in AI engineering [14]. Foundation models act as pre-trained subsystems with limited token windows that constrain reasoning. This bounded working memory propagates upward through the architecture, shaping how context must be selected, modularised, compressed, and loaded at runtime. As GenAI becomes an active collaborator across domains such as education, healthcare, and decision support, humans increasingly co-work with AI [28, 29] on reasoning and decision-making tasks. Yet GenAI systems may produce inaccurate or misleading outputs due to limited contextual awareness and evolving data sources. Architectural mechanisms are therefore needed to govern how persistent knowledge (long-term memory) transitions into bounded context (short-term window) in a traceable, verifiable, and human-aware manner, ensures that human judgment and tacit knowledge remain embedded within the system's evolving context for reasoning and evaluation.

This paper introduces a file-system abstraction as an architectural foundation, as a stepping stone for context engineering, inspired by the Unix philosophy that "everything is a file"[23]. The abstraction provides a persistent, hierarchical, and governed environment where heterogeneous context sources, such as memory, tools, external knowledge, and human contributions, are mounted and accessed uniformly. Building upon this infrastructure, the paper extends file system into a context-engineering pipeline that operationalises context construction under explicit architectural design constraints with the token window. The pipeline performs selection, compression, and incremental streaming of context to ensure that bounded context capacity is used efficiently and transparently.

Section 2 reviews related developments in SE4AI and motivates the need for architectural foundations for context engineering. Section 3 introduces the file system abstraction and its role as a persistent context infrastructure. Section 5 introduces the design constraints and presents the context-engineering pipeline built on these constraints. Section 6 details the AIGNE implementation of the proposed file system abstraction, illustrated through two exemplars. Finally, Section 7 discusses key challenges, future research directions, and concluding remarks.

## 2 Background and Related Work

The emergence of agentic Generative AI systems has given rise to an operating-system paradigm for LLMs (*LLM-as-OS*), which conceptualises the LLM as a kernel orchestrating context, memory, tools, and agents. AIOS project [9] operationalises this paradigm through OS-like primitives for scheduling, resource allocation, and memory management for multi-agent systems [17]. Recent work on further extends this view by proposing an LLM-based semantic file system that enables natural-language–driven file operations and semantic indexing [24] MemGPT [20] introduces a memory hierarchy that coordinates both short-term (context window) and long-term (external storage) memory. While the LLM-as-OS paradigm provides an intuitive high-level conceptual model, it lacks a software-architectural abstraction for how context is structured, shared, and governed. In particular, existing implementations often treat memory, retrieval, and tool use as independent components rather than a coherent infrastructure.

In parallel, context engineering[10, 12] has emerged as a central element in design of Generative AI system. Unlike traditional prompt engineering, which considers context as just a fixed block of text, context engineering treats it as a living, structured mix of instructions, external knowledge, tool definitions, memory, system state, and user queries. Frameworks such as LangChain [13], AutoGen [19] provide partial support through modular components for memory and tool orchestration, but they lack unified mechanisms for traceability, governance, and lifecycle management of context artefacts. Emerging link-based mechanisms [4] treats context as interconnected, discoverable resources, highlighting the need for an unified, verifiable infrastructure to manage such dynamic context. Recent survey [18] from academia also confirms that current approaches are fragmented and identified gaps in verification and lifecycle support. An integrated framework is proposed for bridging context construction and retrieval [6] but note the absence of a verifiable architectural foundation.

Industry and open-source efforts have converged on long-term memory as a critical capability for agentic systems. Existing solutions can be roughly categorized into embedding based solutions, like mem0 [5] and Letta (formerly MemGPT) [1], and Knowledge Graph (KG) based solutions, like Zep/Graphiti [21] and Cognee [16]. However, across these solutions, context governance, access control, and multi-agent sharing remain largely ad hoc. Most frameworks focus on storage and retrieval optimization rather than architectural composability or verifiable traceability.

Beyond architectural paradigms, growing attention has been paid to the dynamics of human–AI co-work, where humans and AI agents jointly perform reasoning, assessment, and decision-making tasks. Recent studies show that combining human judgment with AI inference can enhance performance when tasks require contextual understanding, ethical reasoning, or tacit domain knowledge [2, 25, 30].

The file system abstraction proposed in this paper aligns with the broader LLM-as-OS paradigm. It provides a persistent and governed infrastructure for mounting and managing heterogeneous context resources. By aligning with core software-architectural principles, including modularity, encapsulation, separation of concerns, and traceability, the file system transforms context engineering from ad hoc practice into a systematic, verifiable, and reusable infrastructure. Human roles are directly embed into the context-engineering architecture, ensuring that tacit knowledge and ethical judgment remain integral parts of system reasoning and evaluation.

## 3 File system as Infrastructure for Context

The file system provides the foundational infrastructure that enables systematic context engineering in GenAI systems.Within this environment, agents and human experts function analogously to operating-system processes, performing file-style operations such as reading, writing, and searching on mounted context resources. The file system defines a uniform namespace and a consistent set of basic operations that together enable scalable coordination between autonomous and human actors. The overall architecture of the proposed infrastructure is represented in Figure 1. This architectural abstraction aligns with established software engineering first principles. Concepts such as abstraction, modularity, encapsulation, separation of concerns, and composability shape how context resources are represented, accessed, and evolved. By applying these principles, the file system transforms the complexity of heterogeneous context into a structured, verifiable, and extensible environment for human–AI co-work. The file system operationalises *LLM-as-Operating-System* paradigm by transforming the metaphor into a concrete software architecture design.

### 3.1 Abstraction

The file system implements the SE principle of *abstraction*, providing a uniform interface that hides the heterogeneity of underlying context sources. Regardless of whether a resource is a knowledge graph, a memory store, or a human-curated note, it is represented through a standardised file interface. Because the file system is schema-driven, heterogeneous structures—including REST/OpenAPI resources, GraphQL types, MCP tools, memory stores, or external APIs—can be automatically projected into the namespace. This avoids integration code and turns the file system into a universal semantic interface. This allows agents to reason over diverse context types without knowing their physical format, storage mechanism, or retrieval logic.

### 3.2 Modularity and Encapsulation

The architecture realizes *modularity* by decomposing the environment into independently manageable context resources.

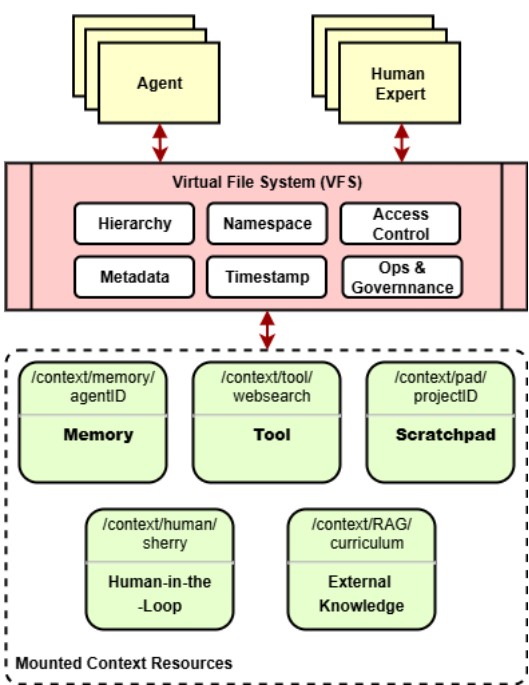

**Figure 1.** File system as a unifying abstraction for context engineering.

Each resource is encapsulated as a mounted component with well-defined boundaries and metadata. This *encapsulation* isolates the internal logic or backend implementation of each resource while exposing only the minimal set of operations required for integration. Thus, changes in one component, for example, swapping a relational database for a vector store, do not propagate across other components in the system. These capabilities eliminate the need to hard-code extensive tool descriptions that would otherwise overload the model's token window. New context sources can be mounted dynamically, similar to Unix file systems, allowing agents to treat external services, tools, or databases as part of a unified addressable space.

### 3.3 Separation of Concerns

Following the SE principle of *separation of concerns*, the file system distinguishes between data, tools, and governance layers. Non-executable files, such as `config.yaml` or `experiment_results.csv`, serve as data or knowledge resources, while executable artefacts such as `analyser.py` or `simulate.sh` represent active tools. This clear distinction ensures that agents and human experts can interpret intent and behaviour correctly, applying appropriate verification and execution strategies. Separation of concerns also extends to governance: access control, log, and metadata management are handled through dedicated mechanisms that remain independent of the functional logic of retrieval or reasoning.

### 3.4 Traceability and Verifiability

Every interaction with the file system, whether initiated by an agent or a human, is logged as a transaction in the persistent context repository. This enforces *traceability* that enables the reconstruction of context provenance and accountability of actions. Coupled with structured metadata, these logs support *verifiability* by allowing changes, reasoning steps, and tool invocations to be audited retrospectively. This ensures that the context pipeline is not only functional but also transparent and auditable.

### 3.5 Composability and Evolvability

The file system achieves *composability* by defining a consistent namespace and interoperable metadata schema across all mounted resources. Context elements can be combined, queried, or integrated into higher-level reasoning processes without additional integration code. *Evolvability* is realised through a plugin architecture that allows new backends, such as full-text indexers, vector databases, or knowledge graphs to be mounted seamlessly without modifying the other components.

Beyond standard file operations, the abstraction can associate each file or directory with meta-defined actions. These actions specify callable behaviours discoverable by agents, ranging from analytical functions, including summarisation, validation, and synchronisation, to domain-specific transformations. Actions elevate each file or directory into an active node, allowing agents to execute tools, transformations or service calls directly through the file system interface.

## 4 Persistent Context Repository: History and Memory Lifecycle

Large language models are inherently stateless: once a session ends, all contextual information is lost. To sustain coherent reasoning across sessions, a GenAI system requires an external, persistent memory repository that captures, structures, and evolves context over time. The *Persistent Context Repository* enabled by the File System fulfills this role. It unifies history, memory, and scratchpad into a continuous lifecycle, ensuring that both short-term and long-term contextual knowledge remain accessible, traceable, and up to date.

When an interaction occurs, raw data are first appended to *History*. *Summarisation*, *embedding* and *indexing* transforms these records into *Memory* representations optimized for retrieval and reasoning. During reasoning, temporary information are written to *Scratchpads*, which may be selectively inserted into *Memory* or archived in *History* after validation. This layered design ensures that all context resources remain both traceable and reusable across agents and sessions.

These components differ in persistence: history is global and permanent; memory is agent-specific or session-specific,

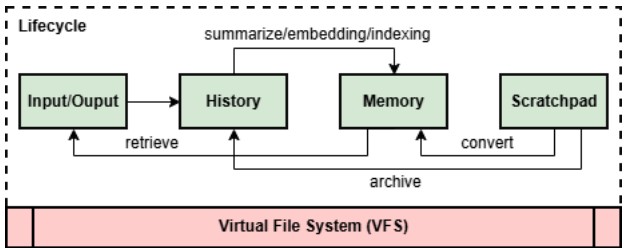

**Figure 2.** Lifecycle of History, Memory and Scratchpad

persistent but mutable; and scratchpads are transient yet auditable. Short-term context, assembled dynamically within the model's token window, functions like working memory. Long-term context, by contrast, must reside outside the model and be selectively included when needed.

The file system enables provenance through timestamps, version control, and access policies. Each transformation, from history to memory, or from scratchpad to memory, is logged as a verifiable state transition. Each artefact carries its creation context, ownership, and lineage, enabling verifiable reconstruction of the reasoning process. This makes the repository not only a data store but also a traceability infrastructure that aligns context management with software engineering principles of modularity and traceability.

### 4.1 History: Immutable Source of Truth

History records all raw interactions between users, agents, and the environment. Each input, output, and intermediate reasoning step is logged immutable and enriched with metadata such as timestamp, origin, and model version. History acts as a verifiable source of truth. It can span multiple agents and sessions, forming a shared global data record accessible through the file system namespace (e.g., */context/history/*). By maintaining complete traces, the system preserves the provenance of reasoning and enables post-hoc analysis, debugging, and compliance verification.

### 4.2 Memory: Structured and Indexed Views

From the context management perspective, memory can be classified along temporal, structural, and representational dimensions.

- Temporal: How long the memory persists.
- Structural: Size or abstraction level of what's stored, token-level, fact-level, or summary-level.
- Representational: How the memory is modelled internally, as raw text, vector embeddings, structured triples, or summaries.

While the short-term versus long-term distinction originates from human cognition, practical GenAI systems manage a spectrum of memory types that balance persistence with dynamism [26], such as episodic memory (task-bounded

summaries) and fact memory (persistent atomic facts). Semantic or induced memory captures higher-level embeddings derived from clustering or summarisation. Each type serves a complementary role within the GenAI reasoning process, ranging from ephemeral reasoning support to enduring knowledge preservation, and is exposed through a consistent namespace hierarchy.

Memory entries are agent-specific and governed through shared metadata and access-control rules. Each memory item maintains a reference to its historical source, ensuring traceability between summarized and original data. Indexed logs and embeddings enable selective recall without re-scanning the entire history, supporting performance and scalability. In the file system, memory is exposed as */context/memory/agentID* and can be extended through plugins such as vector databases or full-text search engines.

### 4.3 Scratchpad: Temporary Workspace

Scratchpads serve as temporary workspaces where agents compose intermediate hypotheses, computations, or drafts during reasoning. Unlike memory, scratchpads are ephemeral and scoped to a specific task or reasoning episode. However, once a session concludes, relevant artefacts may be inserted into memory or appended to history, completing the loop. Scratchpads are represented in the file system as */context/pad/taskID* and are governed by the same metadata and access-control schema as persistent artefacts.

### 4.4 Governance

The lifecycle of history/memory/scratchpad is governed by explicit policies for versioning, aging, and retention. For example, obsolete scratchpads can be pruned automatically, while historical logs may be compressed but never deleted. Such policies ensure that the system remains both scalable and auditable. The persistent context repository operates as a mounted layer within the file system, using its hierarchical namespace and access-control mechanisms. It also serves as the principal data source for the context engineering pipeline, where selected memory and history artefacts are retrieved, compressed, and injected into the context constrained by the token window. All state transitions and transformations are represented as file-level events with timestamps and lineage metadata, enabling replay, audit, and reversible evolution.

## 5 Context Engineering Pipeline

### 5.1 Design Constraints

GenAI model introduce a unique set of architectural design constraints that collectively define the rationale for the design of the context engineering pipeline, fundamentally shape how context is operated. These constraints are intrinsic to the GenAI model layer, and cascade upward through the software architecture, influencing the structure and behavior of higher-level components. Recognizing and

formalizing these design constraints transforms context engineering from ad-hoc prompting practices into a systematic software-architectural discipline.

**5.1.1 Token window.** The token window of GenAI model introduces a hard architectural constraint, which defines the maximum number of tokens that the model can attend to during a single inference pass. This bounded reasoning capacity, determined by model architecture, sets an upper limit on the amount of active context available at runtime (e.g., 128K for GPT 5[1], 200k for Claude Sonnet 4.5[2]). Moreover, as the length of input prompts increases, the computational cost of GenAI models rises significantly due to the quadratic complexity of the self-attention mechanism [8].

Consequently, the context engineering pipeline must curate, compress, and incrementally stream relevant information from the file system into the model's token window. Persistent information in memory must therefore be modularized and hierarchically organized, enabling selective retrieval and incremental refresh. The pipeline manages the temporal coherence of the active window, ensuring that reasoning remains consistent and traceable within bounded context limit. The simplest mitigation is to truncate or summarise large texts, though this inevitably risks information loss [26].

**5.1.2 Statelessness.** GenAI models are inherently stateless, which do not retain conversational history or memory across sessions. This constraint requires external persistent context repository that records, reconstructs, and governs relevant information across interactions. The stateless nature also drives the need for the session memory mechanisms to restore continuity and avoid redundant computation.

However, persisting state externally introduces secondary challenges related to memory growth and redundancy. As conversational or task histories accumulate, semantically similar entries or repeated experiences can increase quickly, degrading retrieval precision and increasing storage cost. To mitigate these effects, context engineering pipeline requires memory deduplication and consolidation strategies that maintain a memory base with minimal redundancy.

**5.1.3 Non-Deterministic and Probabilistic Output.** Because LLMs produce probabilistic outputs conditioned on sampling parameters (e.g., temperature), identical prompts can yield varying responses. From an architectural perspective, this non-determinism introduces challenges for traceability, testing, and verification. It is required that the context engineering pipeline preserves input–output pairs, metadata, and provenance within the file system to support audit, replay, and post-hoc evaluation.

---

[1] https://platform.openai.com/docs/models/gpt-5-chat-latest
[2] https://www.anthropic.com/claude/sonnet

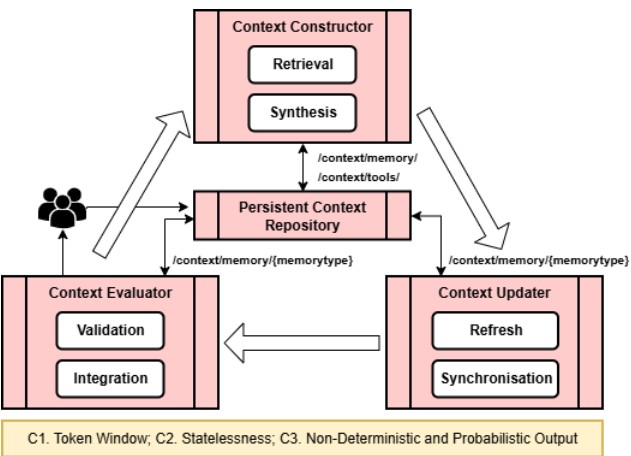

**Figure 3.** The Context Engineering Pipeline.

## 5.2 Design of Context Engineering Pipeline

Building upon the unified architectural foundation established by the file system, this section proposes *Context Engineering Pipeline* that serves as the operational layer that orchestrates context evolution across components.

A pipeline that retains context through both long-term and short-term mechanisms is a key component of an autonomous GenAI agent [7]. Such a pipeline keeps the knowledge and context that is not embedded within the model's weights. The Context Engineering Pipeline bridges context stored in the persistent context repository (history, memory, tools, human input) with bounded reasoning (the token window), ensuring that context is continuously constructed, refreshed, and evaluated throughout the operational lifecycle of an agent. Architecturally, as shown in Figure 3, the pipeline consists of three components: the *Context Constructor*, the *Context Updater*, and the *Context Evaluator*. The architecture operates under three interrelated design constraints as discussed in Section 5.1. The pipeline performs selection, compression, injection, refreshing, and human-in-the-loop evaluation and overwrite, forming a closed loop for context management. Metadata in the file system is used by all the major operations in the context engineering pipeline.

### 5.2.1 Context Constructor.
The Constructor defines how relevant context is selected, prioritized, and compressed from the persistent context repository to prepare bounded, task-specific inputs for reasoning. This process transforms unbounded knowledge into a curated subset that is suitable for the model's active context window. Context selection must also fulfill non-functional qualities such as privacy, access control, and data governance. Because the file system serves as a shared global infrastructure across agents and tasks, the Constructor enforces these constraints to ensure that each reasoning session operates within its authorized scope and the corresponding context remains properly isolated.

Architecturally, the Constructor manages a trade-off between completeness (covering all relevant information) with boundedness (respecting token constraint and cost efficiency). It relies on metadata indicating recency, provenance, which together help infer the relevance of context elements during retrieval and prioritization. Selected context is then compressed through summarization, embedding, or clustering techniques to meet computational budgets, before being aligned with the model's prompt schema [15, 27], a structured input format specifying how context elements are organized for inference.

The Constructor interfaces directly with the file system mount points (e.g., /context/memory/, /context/tool/), queries metadata, and generates a context manifest that records which elements were selected, excluded, and why. This manifest provides transparency, reproducibility, and verifiability for each reasoning session, turning context assembly from an ad hoc operation into a traceable architectural process.

### 5.2.2 Context Updater.
The Context Updater manages the transfer and refresh of constructed context into the bounded reasoning space of the GenAI model. Given the model's limited token window, the Updater must continuously synchronize the token window, the state of the persistent context repository, and the runtime dialogue to maintain coherence and consistency. It ensures that the active context always reflects the most relevant and authorized information, without exceeding model limits or violating access and governance constraints. This synchronization requires continuous monitoring of context size, relevance decay, and temporal and structural dependencies across agents and sessions.

At beginning of the process, a static snapshot of context may be fed into before a single reasoning task for processing. During extended reasoning, incremental streaming allows additional fragments of context to be progressively loaded as the reasoning unfolds. In dynamic or interactive sessions, adaptive refresh mechanisms replace outdated or less relevant fragments in response to model feedback or human intervention. Together, these modes collectively ensure that the reasoning process remains contextually grounded.

All context loading and replacement actions are recorded as metadata events within the file system, including timestamps, source paths, and reasoning identifiers, to enable full traceability and replay of any reasoning session. In multi-agent scenarios, the Context Updater also enforces resource isolation and access separation, ensuring that the context of one reasoning process neither interferes with nor leaks into another.

### 5.2.3 Context Evaluator.
The Context Evaluator closes the loop by verifying model outputs, updating the persistent context repository, and maintaining governance over the evolving knowledge base. It ensures that newly generated

or refined information is validated, contextualized, and reintegrated into the persistent context repository in a traceable and auditable manner.

Model outputs are evaluated against their source context element and provenance metadata to detect hallucinations, contradictions, or context drift. This may involve automated semantic comparison, factual consistency checking, or cross-referencing with authoritative sources. Evaluation metrics, such as confidence scores, factual alignment, and human override rates, are recorded as structured metadata within the file system, supporting post-hoc analysis and traceability.

Verified outputs are transformed into structured memory element, updating or extending the persistent context repository. Long-term memory entries may be appended, revised, or summarized, while episodic memories and scratchpads are pruned or archived. Each update is versioned with timestamps and lineage metadata, ensuring that context evolution remains transparent and reversible.

When confidence thresholds are low or contradictions are detected, the Evaluator triggers human review. Human annotations, ranging from factual corrections to interpretive insights, are stored as explicit context elements, elevating tacit knowledge becomes a first-class component of the knowledge base.

## 6 Implementation Platform: AIGNE Framework

The proposed file system and context engineering pipeline are implemented within the AIGNE Framework[3], a functional development framework designed to simplify and accelerate the creation of GenAI agents. AIGNE provides native integration with multiple mainstream large language models (e.g., OpenAI, Gemini, Claude, DeepSeek, Ollama) and external services via the built-in Model Context Protocol (MCP), enabling dynamic and context-aware application behaviour.

In the AIGNE framework, the AFS (Agentic File System) module serves as the primary file system interface. The SystemFS module implements a virtual file system that provides the following key features.

- Supports list, read, write, and search commands for managing files within mounted directories.
- Enables navigation across nested subdirectories with configurable depth limits.
- Integrates with ripgrep for efficient content search.
- Access to file timestamps, sizes, types, and support user-defined metadata
- Sandboxed access restricted to mounted directories, ensuring isolation and secure file operation

All mounted resources, including MCP modules, memory stores, databases, or external APIs, are projected into the file

system through programmable resolvers. These resolvers implement declarative mappings (similar to GraphQL/OpenAPI schemas) that translate internal structures into AFS nodes without requiring any change to the underlying storage format, enabling semless integration of heterogenous systems.

Within AIGNE, context elements are represented as typed resources under the AFS namespace. Modules such as SystemFS, FSMemory, and UserProfileMemory, each exposing list / read / write / search APIs through standard asynchronous methods. This abstraction enables GenAI agents to access heterogeneous data, like local files, chat histories, and structured memory entries—through a uniform interface without concern for underlying storage backends.

In AIGNE, agents perform reasoning while delegating execution to modular Functions, which are implemented as executable files (e.g., Node.js modules). Each function exports a default asynchronous function together with metadata descriptors (description, input_schema, and output_schema) that allow agents to discover, validate, and invoke them with structured arguments. Functions act as the tools through which agents perform concrete actions, executing code in a sandbox, or calling external APIs.

The Context Constructor is implemented as a process. When a new prompt is received, The constructor executes a series of tool calls such as afs_list() and afs_read(), collecting candidate artefacts (documents, history records, or profile summaries) tagged with metadata including timestamps, provenance, and access scope. The constructor then applies summarisation and token-budget estimation functions to produce a JSON-formatted manifest, which records the selected artefacts, their ordering, and their estimated contribution to the model's prompt. This manifest is passed downstream to the Context Updater.

The Context Updater is realised as part of AIGNE's agent workflow engine. The updater streams context fragments into the model's input buffer during dialogue. In single-turn tasks, it performs a one-off injection of a static snapshot; in interactive sessions, it incrementally refreshes the prompt by invoking AFS read operations to replace or append elements as reasoning unfolds.

The Context Evaluator leverages AIGNE's memory modules to persist newly generated information. After each model response, validated outputs, like summarised user preferences or extracted factual statements, are written back to AFS, stored under directories such as /context/memory/fact/. Each entry is enriched with lineage metadata (createdAt, sourceId, confidence, and revisionId) to support audit and rollback. When the Evaluator detects uncertainty (e.g., confidence below threshold or inconsistent information), it triggers a human-verification stage: annotations are appended as separate artefacts in /context/human/.

---

[3]https://github.com/AIGNE-io/aigne-framework

### 6.1 Exemplar 1: Memory-Enabled Context Construction

AIGNE enables agents to maintain contextual coherence across multiple dialogue turns. Memory is activated declaratively during agent construction through the `DefaultMemory` module, which persists conversation history as retrievable context. The storage location is specified as a file path (e.g., `file:./memory.sqlite3`), allowing memory data to be saved and reloaded across sessions. Each dialogue round is appended to memory and automatically incorporated into subsequent reasoning, enabling long-term, stateful interaction without explicit state management.

```
1  import { AIAgent } from "@aigne/core";
2  import { AFS } from "@aigne/afs";
3  import { AFSHistory } from "@aigne/afs-history";
4  import { UserProfileMemory } from "@aigne/afs-
     user-profile-memory";
5
6  const sharedStorage = { href: "file:./memory.
     sqlite3" }; // use a SQLite database as
     memory storage provider
7
8  const afs = new AFS()
9    .mount(new AFSHistory({ storage: sharedStorage
     }))  // message history memory
10   .mount(new UserProfileMemory({ storage:
     sharedStorage, context: aigne.newContext()
     })); // user memory
11
12 const agent = AIAgent.from({
13   instructions: "You are a friendly chatbot",
14   inputKey: "message",
15   afs,
16 });
```

**Listing 1.** Defining an agent with persistent memory.

### 6.2 Exemplar 2: MCP with Github

The second exemplar demonstrates that any MCP (Model Context Protocol) server can be mounted as an AFS module, exposing its capabilities through a unified file system interface. Using the GitHub MCP server as a real-world case, it shows AI agents interact with GitHub as if they were simply accessing files. Once mounted, the agent can invoke all GitHub MCP tools directly, using `afs_exec` on `/modules/github-mcp/search_repositories` and `/modules/github-mcp/list_issues`.

```
1  import { AIAgent } from "@aigne/core";
2  import { AFS } from "@aigne/afs";
3  import { MCPAgent } from "@aigne/core";
4
5  const mcpAgent = await MCPAgent.from({ // create
     agent from  GitHub official MCP Server
6    command: "docker",
7    args: [
8      "run", "-i", "--rm",
```

```
9      "-e", `GITHUB_PERSONAL_ACCESS_TOKEN=${
       process.env.GITHUB_PERSONAL_ACCESS_TOKEN}`,
10     "ghcr.io/github/github-mcp",
11   ],
12 });
13
14 const afs = new AFS()
15 .mount(mcpAgent);  //Mounted at /modules/github-
     mcp
16
17 const agent = AIAgent.from({
18   instructions: "Help users interact with GitHub
       via the github-mcp-server module.",
19   inputKey: "message",
20   afs,//Agent accesss to all mounted modules});
```

**Listing 2.** Attaching a GitHub MCP function.

## 7 Conclusion and Future Work

Grounded in the emerging *LLM-as-Operating-System* paradigm, this paper presents a file system–based abstraction for context engineering. On this foundation, agents and humans interact as OS-like processes applying standard file operations governed by metadata and transaction logs. The implementation within the AIGNE framework and accompanying exemplars demonstrate the feasibility and adaptability of the proposed approach. Treating context as files further enables GenAI agents to become traceable and auditable, allowing context to be versioned, reviewed, and deployed using DevOps and data-ops practices rather than ad hoc prompt management. By treating the file system as a universal context projection layer, the architecture provides a concrete substrate for emerging LLM-as-OS paradigms, enabling agents to navigate, organize, and evolve their own world models in a verifiable, human-aligned manner.

Future extensions will explore agentic navigation within the AFS hierarchy, enabling agents to autonomously browse, construct indices, and evolve data structures in the mounted space. By allowing agents to function as self-organising processes that observe and modify their own context, the architecture can gradually evolve into a living knowledge fabric, where reasoning, memory, and action converge within a verifiable and extensible file system substrate. Another important direction is to strengthen human–AI co-work, empowering humans not only to oversee or correct system behaviour but also to contribute to, curate, and contextualise knowledge as active participants in context engineering.

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
