# OpenReview forum: "Everything is Context: Agentic File System Abstraction for Context Engineering"
_ACM.org/AIWare/2026/Conference — Submitted to AIware 2026_

### Official Review · Reviewer_oi3Y · 2026-03-03

**Rating:** 2
**Confidence:** 4

**Review:**

Although the idea is very interesting and intuitively should be very practical, the paper does not provide any evidence of the benefits.
The two examples briefly show two components implemented in the AIGNE framework, but there are no comparisons with existing agentic frameworks, to show for example that one can implement these with fewer lines of code, or better separation of concerns, etc.

Since the ideas in this paper are already implemented as a framework (I assume), a small empirical evaluation could show the benefits more clearly.
A recent agentic framework like Mini-SWE-Agent uses only a shell commandline as its interface, and the shell works on a linux system.
This could be a good baseline to compare to, since it can show how the addition of memory, history, etc. as separate files benefits an agent that already works with the filesystem.

**Summary:**

This paper presents the idea of using filesystem as a unified interface for agents.
The paper elaborates how each component or aspect of an agentic workflow maps to a file, and how this abstraction enables managing limited context windows of LLMs, statelessness, etc.

---

> ### Author Response · Authors · 2026-03-13
>
> Thank you for your review feedback and for the thoughtful comments on our submission.
>
> We acknowledge the concern regarding the lack of a more comprehensive evaluation. We fully agree that a rigorous evaluation is important and plan to conduct more systematic studies in the near future. We would like to confirm that the concepts proposed in the paper have been implemented through the AIGINE framework, which serves as a concrete instantiation of the "Everything is Context" design philosophy described in the manuscript.
>
> As evidence of practical use (in addition to evaluation), we would also like to note that the ideas presented in this paper have attracted interest from the broader AI engineering community beyond the academic venue. For example, the work was discussed by several AI practitioners and commentators on social media, including posts by Rohan Paul and Aakash Gupta, both widely followed in the AI engineering community. Rohan Paul’s discussion thread alone received hundreds of engagements (≈350+ likes and dozens of replies), and the paper was subsequently shared and discussed by additional practitioners. While such signals do not replace academic evaluation, they suggest that the concepts introduced in this work resonate with practitioners working on agentic systems and context engineering.

---

> > ### Comment · Reviewer_oi3Y · 2026-03-14
> >
> > Thank you for providing the information about the interest of the community. It is very nice to see such reaction for novel ideas like this.
> >
> > However, without "showing" the benefits of this idea, it would be very hard to convince the community to work on and use this framework.
> >
> > My suggestion for some minimal experiments:
> > - Use the first example in the paper to apply multiple code changes in sequence that each depends on the previous change, and compare an agent in your framework with a standard coding agent.
> > - For the second example, you can define a task that benefits from use of the GitHub MCP, and show that an agent in your framework can perform this task, while vanilla agents have a hard time doing so.
> >
> > These case studies could serve as evidence for the benefits of this idea for this paper, and also for practitioners as some demos.

---

> > > ### Author Response · Authors · 2026-03-15
> > >
> > > Thank you for the constructive suggestion and the excellent idea of using change scenarios for evaluation. Regardless of the outcome of this submission to AIware, we plan to conduct such evaluations and extend the current work in our arXiv version. Your suggestions on sequential code changes and tasks involving GitHub MCP are very helpful. Thank you again for the thoughtful feedback.

---

### Official Review · Reviewer_U5HE · 2026-03-08

**Rating:** 3
**Confidence:** 3

**Review:**

### Strengths

#### 1. I like the main idea of the paper. Borrowing the Unix style “everything is a file” view to organize memory, tools, external knowledge, and human input under one abstraction is quite neat, and it gives the paper a clear systems flavor.

#### 2. The paper is not just presenting an abstract idea. The two cases make the proposal more concrete and show that the design can actually be used in practice.

#### 3. Another nice point is that the paper looks at context engineering from a software architecture perspective, not only as a prompting issue. That makes the discussion more meaningful for agentic systems.


### Weaknesses

#### 1. My main concern is that the paper is still a bit high-level. The two cases are helpful, but they are closer to demonstrations than a solid evaluation, so it is still hard to judge how much improvement the proposed design brings in practice.

#### 2. Also, the presentation can be improved. The figures would be better exported in PDF format instead of images, since some of them may lose clarity after zooming.

**Summary:**

The paper focuses on agentic GenAI systems and highlights context engineering as a central challenge beyond prompt engineering. It introduces an agentic file system abstraction to organize heterogeneous context sources, including memory, tools, external knowledge, and human input, within a unified management framework. Based on this abstraction, the paper further presents a pipeline consisting of a Context Constructor, a Context Updater, and a Context Evaluator. Two case studies in the AIGNE framework demonstrate the effectiveness of the proposed approach.

---

> ### Author Response · Authors · 2026-03-13
>
> Thank you for your positive comments. If you are interested, please also refer to our response to the other reviewers, where we provide additional evidence of practical use and recognition within the AI engineering community.

---

### Official Review · Reviewer_CNAg · 2026-03-12

**Rating:** 2
**Confidence:** 3

**Review:**

originality:
- Architecture may seem interesting, but papers build upon existing framework. Currently, it is whether these architecture is provided by the framework itself, or whether paper introduces new contribution.

presentation:
- Paper is easy to read with minor presentation errors. For example, Section 3 - "systems.Within"

soundness
- In the abstract the motivation behind "everything is a file system" and how this helps context engineering is unclear. Upon reading introduction, it is still unclear why file system architecture aids context engineering.
- the paper describes importance of "unified mechanisms for traceability, governance, and lifecycle management of context artefacts" or "architectural composability", which are mostly interface abstraction, but does not provide any empirical validation. For example, there is no quantitative validation, benchmark comparison or any improvement in results with respect to reasoning accuracy, hallucination, cost and latency.
- The case studies seemed underwhelming; there is no large-scale agent study on real workloads.
- Papers utilizes a lot of fancy terms, like context rot, separation of concerns, improvements in traceability, and human-AI-Co-Work, but as of now, none of these claims can be backed up in terms of how they improve performance, improve security or any other metrics.

**Summary:**

The paper argues that managing context is the main challenge in building Generative AI and agent systems. Instead of focusing only on prompt engineering or retrieval, the authors propose a software architecture for context engineering. The key idea is to treat all context resources as files in a virtual file system, inspired by the Unix principle “everything is a file".

---

> ### Author Response · Authors · 2026-03-13
>
> Thank you for your review feedback and for the thoughtful comments on our submission.
>
> We acknowledge the concern regarding the lack of a more comprehensive empirical evaluation. The primary contribution of this paper is the architectural design of the "Everything is Context" abstraction for agentic systems. We fully agree that a rigorous evaluation is important and plan to conduct more systematic studies in the near future.
>
> We would also like to note that the ideas presented in this paper have attracted interest from the broader AI engineering community beyond the academic venue. For example, the work was discussed by several AI practitioners and commentators on X.com, including posts by Rohan Paul and Aakash Gupta, both widely followed in the AI engineering community. Rohan Paul’s discussion thread alone received hundreds of engagements (≈350+ likes and dozens of replies), and the paper was subsequently shared and discussed by additional practitioners. While such signals do not replace academic evaluation, they suggest that the concepts introduced in this work resonate with practitioners working on agentic systems and context engineering.
>
> We would also like to clarify a potential misunderstanding regarding the relationship between AIGINE/AFS and the paper. The paper does not build upon AIGINE as a prior system; rather, the concepts proposed in the paper have been implemented through the AIGINE system, which serves as a concrete instantiation of the "Everything is Context" design philosophy described in the manuscript.

---

### Official Review · Reviewer_WGAy · 2026-03-12

**Rating:** 2
**Confidence:** 3

**Review:**

Strengths

1. The paper addresses an important problem. Managing context and memory is becoming a central challenge in agentic LLM systems.

2. The file-system analogy is simple and easy to understand at a high level, even though the details of the system were harder to follow.

Weaknesses

1. Unclear core contribution

It is difficult to identify the main technical contribution of the paper. Many of the described ideas—persistent memory, context selection pipelines, and tool integration—already exist in current agent frameworks. The paper would benefit from clearly stating what is new compared to existing systems.

2. No empirical evaluation

The paper does not include experiments or quantitative evaluation. There are no comparisons with existing approaches and no metrics demonstrating improvements in performance, efficiency, or usability. The examples presented are illustrative but not sufficient to validate the proposed architecture.

3. Reads more like a position paper

Much of the paper focuses on high-level architectural ideas rather than concrete technical contributions. Without evaluation or deeper technical analysis, the work feels closer to a position paper than a research paper.

4. Writing quality

I found the paper difficult to follow. Many sections repeat similar high-level ideas and use broad conceptual language instead of clearly explaining the design decisions. This makes it harder to understand what the system actually does and why it is better than existing approaches.

5. Limited discussion of practical trade-offs

The paper does not discuss how the proposed system performs in practice. For example, it would be useful to understand how the approach scales as the number of context artifacts grows, and what the latency or overhead of the file-system abstraction is.

Suggestions for Improvement:

1. Clearly state the main technical contributions and how they differ from existing agent frameworks.
2. Provide empirical evaluation demonstrating the benefits of the approach.
3. Compare the system with existing architectures for context management.
4. Improve the clarity of the writing and reduce repetitive conceptual discussion.

**Summary:**

This paper proposes a file-system abstraction for managing context in agentic LLM systems. The idea is to treat different context sources - such as memory, tools, external data, and human inputs—as files in a unified virtual file system. The system then builds a context engineering pipeline consisting of three components: a Context Constructor, Context Updater, and Context Evaluator. The authors implement this design in the AIGNE framework and provide two examples showing how an agent can use persistent memory and interact with GitHub through MCP.

Overall, the paper argues that representing context as file-like resources can make LLM systems more structured, traceable, and maintainable.

The topic is relevant and the high-level idea is reasonable, but the paper lacks a clear technical contribution and empirical validation. The work would need stronger evaluation and clearer presentation to support its claims.

---

> ### Author Response · Authors · 2026-03-13
>
> Thank you for your review feedback and for the thoughtful comments on our submission.
>
> We acknowledge the concern regarding the lack of a more comprehensive empirical evaluation. The primary contribution of this paper is the architectural design of the "Everything is Context" abstraction for agentic systems. The proposed architecture has been implemented through the AIGINE/AFS framework, which serves as a concrete instantiation of the design principles described in the manuscript. We fully agree that a rigorous evaluation is important and plan to conduct more systematic studies in the near future.
>
> We would also like to note that the ideas presented in this paper have attracted interest from the broader AI engineering community beyond the academic venue. For example, the work was discussed by several AI practitioners and commentators on X.com, including posts by Rohan Paul and Aakash Gupta, both widely followed in the AI engineering community. Rohan Paul’s discussion thread alone received hundreds of engagements (≈350+ likes and dozens of replies), and the paper was subsequently shared and discussed by additional practitioners. While such signals do not replace academic evaluation, they suggest that the concepts introduced in this work resonate with practitioners working on agentic systems and context engineering.